# Detection of Apple Sucrose Concentration Based on Fluorescence Hyperspectral Image System and Machine Learning

**DOI:** 10.3390/foods13223547

**Published:** 2024-11-06

**Authors:** Chunyi Zhan, Hongyi Mao, Rongsheng Fan, Tanggui He, Rui Qing, Wenliang Zhang, Yi Lin, Kunyu Li, Lei Wang, Tie’en Xia, Youli Wu, Zhiliang Kang

**Affiliations:** 1College of Mechanical and Electrical Engineering, Sichuan Agriculture University, Ya’an 625014, China; zhanchunyi@stu.sicau.edu.cn (C.Z.); 19915673580@163.com (H.M.); 2023217014@stu.sicau.edu.cn (R.Q.); 18129290344@163.com (W.Z.); linyi@stu.sicau.edu.cn (Y.L.); likunyu@stu.sicau.edu.cn (K.L.); llwqaq0215@163.com (L.W.); xia_tie_en@163.com (T.X.); 2021217008@stu.sicau.edu.cn (Y.W.); 2Sichuan Jiuzhou Electric Group Co., Ltd., Mianyang 621000, China; frs6668@163.com; 3Meishan Leascend Photovoltaic Technology Co., Ltd., Meishan 620000, China; hetanggui96@163.com; 4Sichuan Intelligent Agriculture Engineering Technology Research Center, Ya’an 625014, China

**Keywords:** apple, sucrose concentration, fluorescence hyperspectral imaging system, machine learning

## Abstract

China ranks first in apple production worldwide, making the assessment of apple quality a critical factor in agriculture. Sucrose concentration (SC) is a key factor influencing the flavor and ripeness of apples, serving as an important quality indicator. Nondestructive SC detection has significant practical value. Currently, SC is mainly measured using handheld refractometers, hydrometers, electronic tongues, and saccharimeter analyses, which are not only time-consuming and labor-intensive but also destructive to the sample. Therefore, a rapid nondestructive method is essential. The fluorescence hyperspectral imaging system (FHIS) is a tool for nondestructive detection. Upon excitation by the fluorescent light source, apples displayed distinct fluorescence characteristics within the 440–530 nm and 680–780 nm wavelength ranges, enabling the FHIS to detect SC. This study used FHIS combined with machine learning (ML) to predict SC at the apple’s equatorial position. Primary features were extracted using variable importance projection (VIP), the successive projection algorithm (SPA), and extreme gradient boosting (XGBoost). Secondary feature extraction was also conducted. Models like gradient boosting decision tree (GBDT), random forest (RF), and LightGBM were used to predict SC. VN-SPA + VIP-LightGBM achieved the highest accuracy, with $R_{p}^{2}$, $RMSE_{p}$, and RPD reaching 0.9074, 0.4656, and 3.2877, respectively. These results underscore the efficacy of FHIS in predicting apple SC, highlighting its potential for application in nondestructive quality assessment within the agricultural sector.

## 1. Introduction

Apple is one of the most widely grown fruits in the world. In 2023, according to the survey results of the National Bureau of Statistics [1], the production of apples was 49.6 million tons in China, which occupied a large market, as shown in Figure 1. With the improvement of people’s living standards, the quality of apples is receiving increasingly more attention from consumers, which is mainly assessed from two aspects: external quality and internal quality. Among them, sucrose concentration (SC) is one of the crucial indicators of internal quality [2]. In comparison to visual characteristics, SC often provides a more direct reflection of the taste and flavor of apples [3]. At the same time, SC is one of the significant metrics of quality classification, and achieving detection of the SC in apples becomes especially important.

Traditional testing methods primarily include handheld refractometers, hydrometers, electronic tongues, and saccharimeter analyses [4,5,6]. However, these methods will cause damage to apples to some extent and cannot meet the needs of large-scale nondestructive testing. Therefore, there is an urgent requirement for a nondestructive and rapid method to measure the SC in apples accurately, aiming to overcome the shortcomings associated with traditional testing methods, including potential damage and inconvenience. In recent years, due to the characteristics of rapid and nondestructive detection, spectral technology has been gradually applied to the detection of agricultural products and livestock raising [7]. Yang et al. [8] predicted the water-holding capacity of fresh chicken breast meat by fusion of hyperspectral imaging spectral and textural data, resulting in a predictive correlation coefficient $\left(R_{p}\right)$ of 0.80 and root mean square error (RMSE) of 0.80; the results of the study demonstrated the potential of hyperspectral imaging for the assessment of the water-holding capacity of chicken meat. Hu et al. [9] used hyperspectral imaging to differentiate Tibetan tea grades with a 100% correct classification rate, which indicated that hyperspectral imaging can be used as an alternative method for rapid, nondestructive testing of tea quality. Qi et al. [10] established a regression model that effectively predicted the chlorophyll content of peanut leaves, based on the spectral index, and the root mean square error values were all lower than 2.04, which showed the effectiveness of spectral-based detection of chlorophyll. Shang et al. [11] detected different degrees of adulteration of tea seed oil through hyperspectral technology combined with a partial least squares regression prediction model. Sun et al. [12] used the spectral technique for the detection of pesticide residues in lettuce leaves, and the predicted results had a coefficient of determination of the prediction set $\left(R_{p}^{2}\right)$ of 0.9386 and the root mean square error of the prediction set $\left(RMSE_{p}\right)$ was 0.0077. The application of these spectroscopic techniques provides a powerful reference for agricultural and livestock raising testing.

With the advancement of spectral imaging technology, a fluorescence hyperspectral imaging system (FHIS) consisting of fluorescence spectroscopy and hyperspectral imaging has been developed [13]. This technique is more suitable for multiple scene detection applications due to its multi-band, high-resolution integration [14]. Many studies have demonstrated that the FHIS is not only widely used in industry [15], food [16], and medicine [17] but also plays a role in fruit quality detection. Fu et al. [18] detected early pear injuries and demonstrated the capability to distinguish between varying degrees of bruising as soon as 15 min post-injury with an impressive accuracy of 93.33%. Kang et al. [19] investigated the dry matter content of mango with an $R_{p}^{2}$ = 0.9658. Wang [14] investigated the pH of kiwifruit, with an $R_{p}^{2}$, coefficient of determination of the training set ($R_{c}^{2}),$ and residual predictive deviation (RPD) of 0.8512, 0.8580, and 2.66, respectively.

To the best of our knowledge, there is a dearth of research on apple SC studies leveraging FHISs. Nevertheless, several studies have demonstrated the feasibility of using a FHIS for detection purposes, prompting us to integrate a FHIS into our investigation of apple SC. This study introduced a novel perspective in the field of nondestructive apple quality assessment by investigating the potential of combining the FHIS with machine learning (ML) to detect SC in apples. Firstly, the spectral curves of the equatorial region of apples were obtained through the FHIS. Then, the spectral curve of the equatorial region was optimized by selecting the best preprocessing method among four preprocessing methods. The primary feature selection and the secondary feature selection were chosen to improve the prediction accuracy of the model. And a comparative analysis of three prediction models was carried out to determine the best optimized SC prediction model. In addition, an attempt was made to investigate the identification of spectral feature variables for predicting SC to explain the improvement in predictive performance by feature selection.

## 2. Materials and Methods

### 2.1. Apples Samples

In September 2023, 160 samples of ripe apples with similar sizes (approx. 80 mm in diameter) and no apparent surface defects were picked in Hanyuan as the measurement samples. All apples were cleaned and wiped with a towel and numbered from 1 to 160. In order to eliminate the influence of temperature on the accuracy of the prediction model, the samples were placed in a constant temperature and humidity storage cabinet for 24 h, with a temperature of 25 °C and humidity ranging from 56% to 58% RH.

### 2.2. Fluorescence Hyperspectral Imaging System

The fluorescence hyperspectral images of apple samples were taken by a GaiaFluo (/Pro)-VN-HR series fluorescence hyperspectral test system manufactured by Jiangsu Dualix Spectral Image Technology Co., Ltd. (Wuxi, China) [20]. The system mainly consists of a hyperspectral camera, a camera obscura, a xenon lamp, four reflected light sources, a xenon lamp power supply, and filters, as shown in Figure 2. The equipment comprises two types of filters: excitation and emission filters [21]. The excitation filters are categorized into five bands at wavelengths of 357 nm, 390 nm, 452 nm, 534 nm, and 628 nm, which block light input from other wavelengths, thus minimizing signal interference [22]. The emission filters are added to separate the fluorescence signal from the stray light, enabling the hyperspectral camera to capture the valid fluorescence signal generated by the samples. There are five types of emission filters, each corresponding to wavelengths of 475 nm, 495 nm, 530 nm, 570 nm, and 610 nm. The accurate selection of filters is paramount prior to conducting fluorescence hyperspectral image acquisition; the rational combination of excitation and emission filters at different wavelengths helps to produce the good fluorescence images.

### 2.3. Fluorescence Hyperspectral Data Acquisition

According to the system manual, 390 nm excitation filter is currently the most used excitation filter in fruit detection research, as it is better at cutting off other wavelengths of the input filter [23]. Since the fluorescent substances enriched in apples were excited by the excitation light source and emit high-intensity fluorescence in the range of 440–780 nm, the 475 nm emission filter was chosen so that as many bands of the fluorescence image as possible could be captured. Therefore, the combination of the 390 nm excitation filter and 475 nm emission filter was finally chosen to acquire fluorescence hyperspectral images. After matching the filters, the apple was fixed in a camera obscura by using a foam ring and placed on the sample stage with its equatorial region facing toward the camera position and the direction of the light source. This approach could minimize data fluctuations caused by apple rolling and ensure the accuracy of experimental measurements. The parameters of the spectral acquisition were set in SpecView (version 3.1) software: scanning speed of 0.013 cm/s, exposure time of 800 ms, and RGB values of 638, 551, and 442. The FHIS control and image acquisition were realized in the motion control interface of the SpecView software. The equatorial region of the apple fluorescence spectrum image was manually extracted as the region of interest (ROI) by using ENVI 5.3 software (ITT Visual Information Solutions, Boulder, CO, USA). The average fluorescence hyperspectrum of all pixel points in the ROI was taken as the fluorescence hyperspectrum information of the apple sample, and the extracted data could represent the whole apple sample and effectively reflected the sample information. Each ROI was a circle of the same size and location, and the average spectral data at the ROI was taken as the spectral data for the samples. Thus, the original spectral data of 125 variables ranging from 376.80 nm to 1011.05 nm were obtained.

### 2.4. Acquisition of SC Data

Destructive SC measurements of apple samples were performed using a handheld sugar refractometer (Atago, PAL-1, Tokyo, Japan), which was the common approach to test SC [24,25]. The region identical to the ROI was selected, and the pulp was extracted by removing the peel from the corresponding area of the apple, manually pressing using a juicer, and filtering through gauze. Two to three drops (approx. 0.3 mL) of the apple juice were added to the detection area by a burette. Data recording began after the measurements were stable. In order to avoid the effect of experimental random errors on the accuracy of the results, five sets of identical measurements were carried out and the average of the results was taken as the SC of the apple.

### 2.5. Data Preprocessing

The original fluorescence hyperspectral data contained both sample-related information and interference with irrelevant information, such as camera dark current, light scattering, and noise [26]. It is necessary to preprocess the spectral data to prevent the raw data from affecting the accuracy of the modeling. Data preprocessing is an important step, which aims to improve data quality and improve the accuracy of the model. Standard normal variate (SNV) can make the sample data relocate to a normal distribution and ensure a large amount of information from the sample to be clustered in the center of the sample. A Savitzky–Golay smoothing filter (SG) can improve data accuracy without changing the signal trend. Wavelet transform (WT) can decompose the original spectral data, remove the interference and noise signal in the wavelet coefficient by using the threshold method, and obtain the spectral data after noise removal. Vector normalization (VN) can enhance the inter-sample variance by calculating the mean spectra to minimize and eliminate the effect of interfering information on the spectra. All the preprocessing methods were performed using Python 3.10 (Python Software Foundation, Wilmington, DE, USA).

### 2.6. Feature Extraction Methods

The preprocessed spectral data were of high dimensionality, although a great deal of the interfering information had been removed from them. However, some features in the data were interrelated with the prediction of apple SC, while others were not, as irrelevant information can affect the accuracy of modeling. Feature extraction extracts the sample set from the high-dimensional feature space to the low-dimensional feature space by extracting the relevant information to obtain features that are more predictive of apple SC. Thus, relevant information can be extracted from the spectral data by feature extraction for better prediction of apple SC [27]. In this study, three primary feature extraction methods and their combinations (secondar feature extraction) were used. The primary feature extraction methods include variable importance projection (VIP) [28], a successive projection algorithm (SPA) [29], and extreme gradient boosting (XGBoost) [30]. VIP is an algorithm that calculates an importance score from each feature and uses the most important features for modeling. SPA is a forward-loop selection method, which can reduce the collinearity between variables. XGBoost can effectively calculate feature importance by evaluating the gain of features in tree node splits. This functionality facilitates the rapid identification of the features that are crucial to the model’s performance.

### 2.7. Machine Learning Model

ML is a powerful technology for analyzing data and is widely used in many fields [31]. It trains the model through the input of massive training data so that the model can master the potential rules contained in the data and then make accurate predictions of the newly input data. To evaluate the accuracy of models in predicting apple SC, three kinds of ML regression models, including gradient boosting decision tree (GBDT), random forest (RF), and light gradient boosting machine (LightGBM), were used to model the variables in this study. GBDT can reduce residuals by adding new trees and establishing new models in the direction of residual reduction (negative gradient) [32]. RF can ‘pull the logic’ from the data by providing insights into feature importance and parameter relationships. Individual ‘trees’ are extracted from the model to provide useful guidance [33]. LightGBM has excellent capability and accuracy in data regression, which is achieved by combining exclusive feature bundling and gradient-based unilateral sampling [34].

### 2.8. Model Evaluation

To assess the validity of the SC model predictions, the coefficient of determination ($R^{2}$) and RMSE were employed to evaluate the accuracy of the models [35]. The larger the $R^{2}$, the better the data fit is, while the smaller the RMSE, the smaller the error between the prediction model result and the true value. The calculation formulas were as follows:(1)$$R^{2}=1-\frac{\sum_{i=1}^{n}{\left({\hat{y}}_{i}-y_{i}\right)}^{2}}{\sum_{i=1}^{n}{\left({\hat{y}}_{i}-{\bar{y}}_{i}\right)}^{2}}$$
(2)$$RMSE=\sqrt{\frac{\sum_{i=1}^{n}{\left({\hat{y}}_{i}-y_{i}\right)}^{2}}{n}}$$

In Equations (1) and (2), ${\hat{y}}_{i}$ and $y_{i}$ are the actual and predicted values of the samples, respectively; ${\bar{y}}_{i}$ is the mean value of the samples; and n is the number of samples. In the subsequent model evaluation, $R^{2}$ represents the coefficient of determination, $R_{c}^{2}$ represents the coefficient of determination of the training set, and $R_{p}^{2}$ represents the coefficient of determination of the prediction set. RMSE represents the root mean square error, $RMSE_{c}$ represents the root mean square error of the training set, and $RMSE_{p}$ represents the root mean square error of the prediction set.

RPD was employed to evaluate the models. The calculation formula was as follows:(3)$$RPD=\frac{SD}{RMSE_{p}}$$
where *SD* is the standard deviation of the test values for all samples in the prediction set. When the RPD < 1, it indicates that the model performance is poor and cannot be applied effectively. When the 1 < RPD < 2, it means the model is working well and may be used in the analysis, while RPD > 2 indicates excellent model performance.

## 3. Results and Discussion

### 3.1. Sample Division

Spectral data outliers refer to values that clearly differ from other data points in spectral analysis. They may be caused by noise during data acquisition, instrument malfunction, contamination, sample handling, and other factors. Eliminating outliers in spectral data can improve the quality of the spectral data and the accuracy of the analysis results. The 160 original samples were tested for outliers by using the Monlocat method, and 155 data samples were used as the sample sets after the removal of outliers. The samples were divided into training and prediction sets using the sample set partitioning using joint X-Y distances (SPXY). This method considered the distribution of spectral data and the SC of apples when dividing the samples, making the partition data more reliable. In this study, 155 samples were divided into training and prediction sets at a ratio of 3:1 based on the SPXY. The training set samples were used for model training, while the prediction set samples were used for model performance verification. The results showed that the SC of the training set ranged from 9.8 Brix% to 14.5 Brix%, and the SC of the prediction set ranged from 10.0 Brix% to 14.2 Brix%, as shown in Table 1. Since the data range of the samples in the prediction set was entirely within the data range of the training set, the effectiveness of the data segmentation could be ensured by using SPXY.

### 3.2. Spectrum Analysis

When a xenon lamp light source excites the fluorescent substances in apples, energy will be released during molecular jumps and molecules that have released their energy will return to the ground state, thus resulting in fluorescence. The curve of fluorescence hyperspectral images after ROI extraction by ENVI 5.3 showed different typical characteristic emission peaks, as shown in Figure 3b. The study demonstrated that the original spectrum of apple samples exhibited three distinct emission peaks. These were primarily observed in three wavelength ranges: blue–green light band (440–530 nm), red band (680 nm), and far-red light band (735–780 nm). The presence of carotenoids and chlorophylls in the apple sample’s epidermis accounted for these emission peaks. Specifically, the emission in the 440–530 nm range was influenced by carotenoids and chlorophylls, while the 680–780 nm range was predominantly affected by chlorophylls [36]. Since the emission peaks of chlorophyll-a and chlorophyll-b were different, a double emission peak appears at 680–780 nm. The different peak intensities of the different spectral curves were due to the different levels of carotenoid and chlorophyll components contained in each apple.

### 3.3. Preprocessing Results

In order to eliminate noise interference during the image acquisition process, SNV, SG, WT, and VN methods were employed to preprocess the original spectral data in this study. The spectral curves processed by the four methods were shown in Figure 4. The results indicated that the trend of the processed spectral curves was the same as that of the original spectral curves. The spectral profiles modified by SG and WT maintained the characteristics of the original spectra and the spectral profiles were smoothed. However, SNV and VN had changed considerably from the original data due to the fact that SNV aimed to remove bias and stretching between samples so that the data had a mean of 0 and a standard deviation of 1 in each feature dimension, while VN aimed to adjust the scale or range of the data vectors.

The SC prediction model was built based on three ML models, and the results are shown in Table 2. In GBDT, the $R_{p}^{2}$ was 0.6523, $RMSE_{p}$ was 1.1435, and RPD was 1.3148 in the model without preprocessing. The $R_{p}^{2}$ ranged from 0.6601 to 0.7584, $RMSE_{p}$ ranged from 1.0741 to 1.1306, and RPD ranged from 1.3778 to 1.4329 in the models after four types of preprocessing methods. Compared with the GBDT model without preprocessing and after four types of preprocessing, in all the models after preprocessing, the $R_{p}^{2}$ was boosted, representing an increase in the accuracy of the model in fitting the apple SC, and the $RMSE_{p}$ decreased, representing a decrease in the error of the model in fitting the apple SC. The combined performance was shown by the rise in RPD, which represented the performance improvement of the model. All four preprocessing methods obtained superior performance, indicating that preprocessing indeed provided the effect of improving the prediction results of the GBDT model. But in both RF and LightGBM, the RPD of the WT preprocessed model was lower than in models without preprocessing. Thus, different preprocessing methods had different effects in different models.

By comprehensively comparing the four preprocessing methods, the results obtained by the preprocessing methods of SNV, SG, and WT were all inferior to those of the VN method in three ML models. However, the difference in the $R^{2}$ between the training set and the prediction set was still significant in all the models after VN. This indicated that there was overfitting of the model and continued optimization is needed. Among the three models, the VN method gave the best performance, which indicated the solvable nature of VN, so the data preprocessed by the VN method were chosen for optimization.

### 3.4. Extraction of Effective Variables and Modeling

#### 3.4.1. The Model Prediction Results of the Primary Feature Extraction

By performing feature extraction on the data processed by the VN method, the distribution of its variables was obtained, as shown in Figure 5. A total of 38, 35, and 80 variables were extracted by VIP, SPA, and XGBoost, which represent 30%, 28%, and 64% of the original variables (125 variables), respectively. In the three primary feature extraction models, SPA and VIP extracted almost equal amounts of bands, which were mainly distributed around 600–800 nm. While XGBoost extracted the most bands and had the most even distribution bands, this may be due to the fact that the XGBoost algorithm calculates feature importance in such a way that many features in the spectrum are beneficial for improving the split-point performance, so many of them were selected by the boosted tree [37].

All variables and the three primary feature extraction variables were modeled by using three ML regression models, as shown in Table 3. For the same feature extraction method, there were differences in the performance results of the three models, with GBDT performing the worst compared to the other two, which may be due to the fact that the number of features after feature extraction was still high and GBDT was not good at dealing with data with many dimensions, resulting in a poor prediction performance, whereas RF and LightGBM were able to maintain a better performance for data with high dimensions. Among them, XGBoost-GBDT, VIP-RF, and VIP-LightGBM were the best prediction models with primary feature extraction. By comparing the results of models without feature extraction to those with optimal primary feature extraction, it was clear that the RPD had improved, which meaned prediction performance was improved. This was due to the fact that the primary feature selection method selects more relevant feature variables as model inputs, thus reducing data redundancy. Comparing the three best prediction models with primary feature extraction, the difference between $R_{c}^{2}$ and $R_{p}^{2}$ was about 0.1, which meant that feature extraction could be continued to reduce overfitting.

#### 3.4.2. The Model Prediction Results of the Secondary Feature Extraction

Compared to the number of 125 original variables, the number of feature variables obtained by the above three primary feature extraction methods was greatly reduced, ranging from 28% to 64% of the original variables. The presence of overfitting remained, although the number of variables was reduced. In order to solve this issue, secondary feature extraction was performed using the VIP + XGBoost, the SPA + VIP, and the SPA + XGBoost methods to illustrate the distribution of variables after secondary feature extraction, and the number of variables was significantly reduced to 8~34%. As can be seen in Figure 6, the bands after secondary feature extraction were more censored, and this combination of overlays could be used to change the data dimensions, compensating for the disadvantage of more dimensionality in the primary feature extraction.

Compared to the results of the primary feature extraction, it could be observed that the overall prediction results of secondary feature extraction were generally better, as shown in Table 4. The RPD values of the primary feature extraction for GBDT ranged from 2.8588 to 2.9749, while the RF was 2.8727–3.0256 and the LightGBM was 2.9628–3.2877. The $R^{2}$ of the training and prediction sets could be minimized to a difference of 0.02. This may be due to the fact that secondary feature extraction removed many irrelevant variables and retained the key variables, allowing the model to capture more critical information and reduce interference when predicting, resulting in better predictions. Therefore, it was particularly valuable to perform feature extraction on the variables again after the primary feature extraction. SPA + XGBoost did not perform well compared to the other two secondary feature extraction methods. The results showed that the SPA + XGBoost features selected the least variables; this may be due to the fact that there were too few variables after SPA + XGBoost feature extraction, and the useful information obtained by the model reduced, resulting in a poor prediction effect of the model. In summary, secondary feature extraction could improve the generalization of the model to a certain extent, but a few numbers of the useful information variables could also lead to lower generalization. Thus, the accurate selection of the number of variables with useful information was crucial to ensure the accuracy of model prediction.

### 3.5. Best Predicted Results of the Three Regression Models

Based on the above results, the best feature extraction methods of the three regression models were determined to be the SPA + VIP-GBDT, the VIP + XGBoost-RF, and the SPA + VIP-LightGBM. The model parameters are shown in Table 5, and the comparison of the best results of the three models is shown in Table 6.

The fitting results of the three best ML models, SPA + VIP-GBDT, VIP + XGBoost-RF and SPA + VIP-LightGBM, for the apple SC were calculated, as shown in Figure 7. The feature extraction method SPA + VIP had a good performance in both GBDT and LightGBM. Overall, the spectra that were preprocessed by VN produced good results, but some of the preprocessing methods produced bad results in all three ML models. This may be because this part of the preprocessing reduced the difference in the features of the spectra. Primary feature extraction and secondary feature extraction reduced many redundant variables and effectively avoided the disadvantage of overfitting by selecting the features that were highly correlated with the target variables. This resulted in improved accuracy and performance of the apple SC model. Among them, the best prediction model among the three prediction models was LightGBM, and the best method for feature extraction was SPA + VIP. The values of $R_{p}^{2}$, $RMSE_{p},$ and RPD were 0.9074, 0.4656, and 3.2877, respectively.

### 3.6. Identification of Feature Variables and Correlation Analysis

In order to better understand the reason why feature extraction could improve the prediction performance, the performance effect in different models after different feature extraction methods was further analyzed. As can be seen from Table 6, the bands after SPA + VIP feature extraction performed well in GBDT and LightGBM, and the best performance was in SPA + VIP-LightGBM. Further observation of the 10 bands after SPA + VIP feature extraction showed that the extracted bands were mainly located in the second and third wave crests. They were at 660 nm, 665 nm, 670 nm, 680 nm, 711 nm, 742 nm, 778 nm, 783 nm, and 794 nm, respectively. The wave peaks of sucrose contained in apples at 680 nm were mainly related to the multiplicative frequency expansion vibration of the C-H and O-H bonds, the peaks of the wave peaks at 800 nm were mainly related to the C-H bond and N-H bond, the wave peak at 800 nm was mainly related to the C-H bond and N-H bond secondary octave absorption, and the wave crest near 750 nm was caused by the triple-frequency stretching vibration of O-H [38,39]. These bands were correlated with the SC, so these could be used as characteristic variables to predict the SC of apples as well.

## 4. Conclusions

In this study, a FHIS combined with ML was used to detect SC in apples. When apples were excited by a fluorescent light source, they exhibited distinctive fluorescence features at 440–530 nm and 680–780 nm, allowing the FHIS to predict SC. The VN preprocessing method improved the prediction ability compared to the original spectra. VIP, SPA, XGBoost, and their combined feature variable extraction methods not only reduced the number of variables input into the model but also further improved the prediction ability. It was shown that the secondary feature extraction methods outperformed the modeling results using raw spectral data and primary feature extraction in predicting apple SC. Among them, the model demonstrated the best performance in predicting SC, with the $R_{p}^{2}$, $RMSE_{p}$, and RPD of 0.9074, 0.4656, and 3.2877. Compared with the original spectra, $R_{p}^{2}$ was improved by 0.2102, $RMSE_{p}$ was reduced by 0.6014, and RPD was improved by 1.4293. In conclusion, the advantage of spectra that will be used after feature extraction is that, through feature extraction, the model can extract more relevant and useful information, thus improving the prediction of apple SC. The performance of this model outperformed the results of the multispectral features constructed by Tang et al. to predict apple SC ($R_{p}^{2}$ = 0.8861, $RMSE_{p}$ = 0.8738) [40] and also outperformed the results of the spatially distributed spectral-based prediction of peach SC by Huang et al. ($R_{p}^{2}$ = 0.853, *RPD* = 1.6) [41]. This paper was the first to use a FHIS to study apple SC and has used multiple feature extraction methods to confirm its advantages. The final prediction model RPD reached 3.2877, indicating that the model has the ability to accurately predict the apple SC, with the model results superior to the multispectral and differential spectroscopy mentioned in this study.

In summary, the results showed that it was feasible to nondestructively detect the SC of apples by using a FHIS with ML. This approach will also provide an experimental and theoretical basis for rapid, nondestructive, and high-precision detection of fruit quality.

## Figures and Tables

**Figure 1 foods-13-03547-f001:**
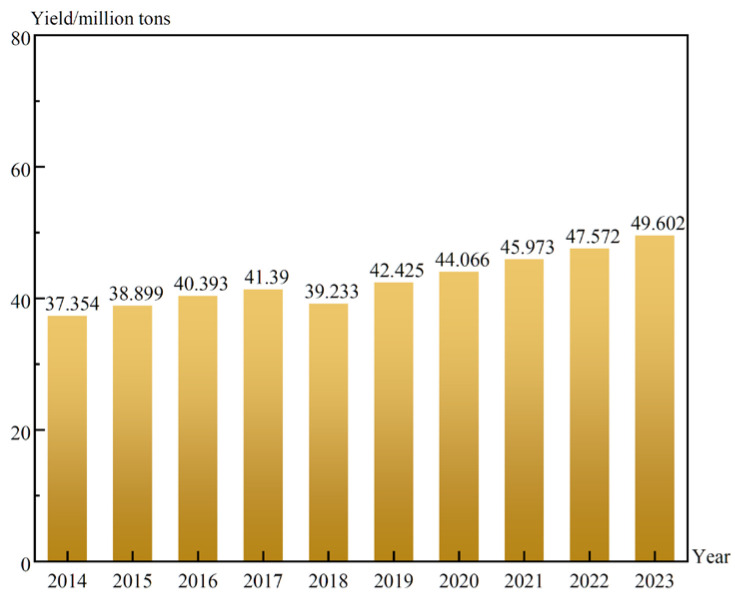
China’s apple production in the last ten years.

**Figure 2 foods-13-03547-f002:**
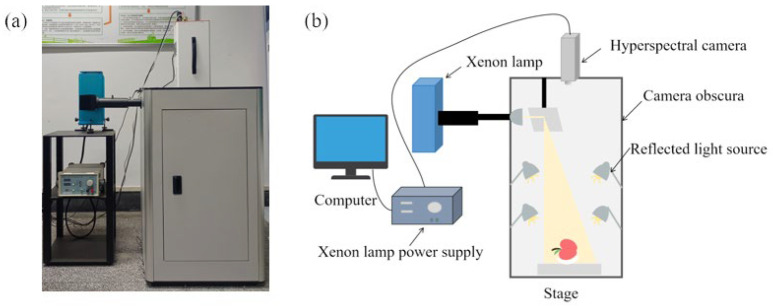
FHIS: (**a**) data acquisition platform; (**b**) the schematic diagram.

**Figure 3 foods-13-03547-f003:**
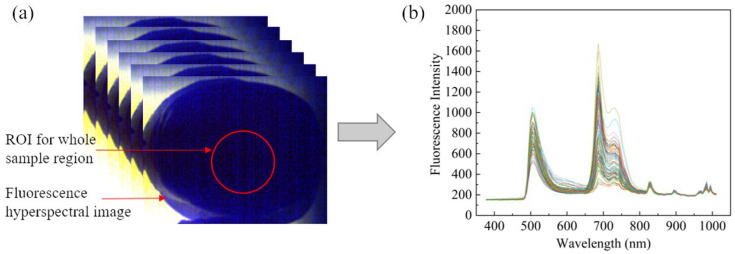
Spectral curve acquisition: (**a**) ROI extraction; (**b**) the original fluorescence hyperspectral curve.

**Figure 4 foods-13-03547-f004:**
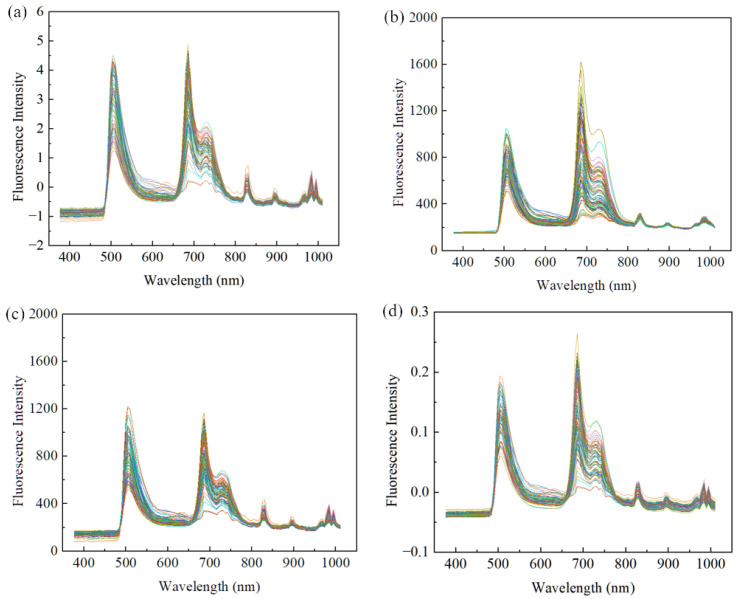
The spectra after different preprocessing methods: (**a**) SNV, (**b**) SG, (**c**) WT, (**d**) VN.

**Figure 5 foods-13-03547-f005:**
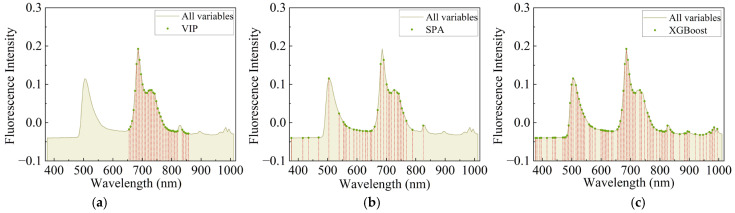
Distribution of variables after primary feature extraction: (**a**) VIP, (**b**) SPA, (**c**) XGBoost.

**Figure 6 foods-13-03547-f006:**
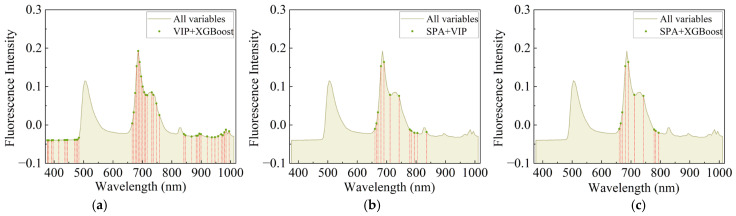
Distribution of variables after secondary feature extraction: (**a**) VIP + XGBoost, (**b**) SPA + VIP, (**c**) SPA + XGBoost.

**Figure 7 foods-13-03547-f007:**
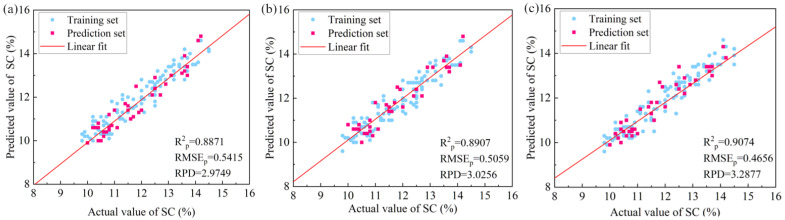
The plot of predictive fit of the three best regression models: (**a**) SPA + VIP-GBDT, (**b**) VIP + XGBoost-RF, (**c**) SPA + VIP-LightGBM.

**Table 1 foods-13-03547-t001:** SC statistics and sample division parameters based on SPXY.

Sample Set	Number of Samples	Minimum Value (Brix%)	Average Value (Brix%)	Max Value (Brix%)	Standard Deviation
Training set	117	9.8	11.8	14.5	1.26
Prediction set	38	10.0	11.7	14.2	1.25
Total	155	9.8	11.8	14.5	1.26

**Table 2 foods-13-03547-t002:** Prediction results of SC using different preprocessing methods.

Model	Preprocessing Method	Training Set		Prediction Set		RPD
		$R_{c}^{2}$	*RMSE* _c_	$R_{p}^{2}$	*RMSE_p_*	
GBDT	None	0.8660	0.9448	0.6523	1.1435	1.3148
	SNV	0.8498	0.9918	0.7131	1.0964	1.3815
	SG	0.8612	0.9613	0.6674	1.1184	1.3778
	WT	0.8713	0.9260	0.6601	1.1306	1.4165
	VN	0.8547	0.9778	0.7584	1.0741	1.4329
RF	None	0.8736	0.9175	0.7075	1.0489	1.9025
	SNV	0.8394	1.0255	0.7391	1.1756	1.9721
	SG	0.8735	0.9179	0.7147	1.0358	2.0259
	WT	0.8611	0.9618	0.6530	1.1423	1.6964
	VN	0.8579	0.9671	0.7897	1.0019	2.0500
LightGBM	None	0.9074	0.7852	0.6972	1.0670	1.8584
	SNV	0.8894	0.8510	0.7683	1.1078	2.0342
	SG	0.9087	0.7798	0.6961	1.0691	1.8784
	WT	0.9096	0.7758	0.7061	1.0513	1.8275
	VN	0.8943	0.8341	0.8053	0.9640	2.1321

**Table 3 foods-13-03547-t003:** The model prediction results of the primary feature extraction.

Model	Primary Feature Extraction Method	Training Set		Prediction Set		RPD
		$R_{c}^{2}$	*RMSE* _c_	$R_{p}^{2}$	*RMSE_p_*	
GBDT	None	0.8547	0.9778	0.7584	1.0741	1.4329
	VIP	0.8376	1.0055	0.7321	1.0555	1.4345
	SPA	0.8460	1.0192	0.7266	1.1206	1.3718
	XGBoost	0.8563	0.9724	0.7436	1.1057	1.4710
RF	None	0.8579	0.9671	0.7897	1.0019	2.0500
	VIP	0.8616	0.9642	0.7654	0.9878	2.1225
	SPA	0.8762	0.9141	0.7694	1.0290	1.8989
	XGBoost	0.8047	1.1337	0.7302	1.1343	2.1221
LightGBM	None	0.8943	0.8341	0.8053	0.9640	2.1321
	VIP	0.8837	0.8837	0.8071	0.8956	2.2248
	SPA	0.8698	0.9373	0.8046	0.9473	2.1254
	XGBoost	0.8990	0.8153	0.7879	1.0058	2.1148

**Table 4 foods-13-03547-t004:** The model prediction results of the secondary feature extraction.

Model	Secondary Feature Extraction Method	Training Set		Prediction Set		RPD
		$R_{c}^{2}$	*RMSE* _c_	$R_{p}^{2}$	*RMSE_p_*	
GBDT	VIP + XGBoost	0.8923	0.4399	0.8834	0.5444	2.8799
	SPA + VIP	0.9028	0.4916	0.8871	0.5145	2.9749
	SPA + XGBoost	0.9062	0.5952	0.8867	0.5512	2.8588
RF	VIP + XGBoost	0.9001	0.4987	0.8907	0.5059	3.0256
	SPA + VIP	0.9009	0.4801	0.8975	0.5119	3.0024
	SPA + XGBoost	0.8926	0.9626	0.8873	0.5201	2.8727
LightGBM	VIP + XGBoost	0.9066	0.4349	0.8955	0.4543	3.1051
	SPA + VIP	0.9102	0.4727	0.9074	0.4656	3.2877
	SPA + XGBoost	0.8945	0.6153	0.8825	0.5404	2.9628

**Table 5 foods-13-03547-t005:** The parameter settings of the three models.

Model	Method	Parameter Settings			
		N-Estimators	Max-Depth	Learning-Rate	Min-Child-Samples
GBDT	SPA + VIP	90	4	0.02	3
RF	VIP + XGBoost	10	4	—	—
LightGBM	SPA + VIP	45	2	0.1	5

**Table 6 foods-13-03547-t006:** Comparison of the optimal results of the three models.

Model	Secondary Feature Extraction Method	Training Set		Prediction Set		RPD
		$R_{c}^{2}$	*RMSE* _c_	$R_{p}^{2}$	RMSE_p_	
GBDT	SPA + VIP	0.9028	0.4916	0.8871	0.5145	2.9749
RF	VIP + XGBoost	0.9001	0.4987	0.8907	0.5059	3.0256
LightGBM	SPA + VIP	0.9102	0.4727	0.9074	0.4656	3.2877

## Data Availability

The original contributions presented in the study are included in the article, further inquiries can be directed to the corresponding author.

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
