# Peer review of "Detection of Apple Sucrose Concentration Based on Fluorescence Hyperspectral Image System and Machine Learning"

_foods, 2024, doi:10.3390/foods13223547_

Round 1
Reviewer 1 Report
Comments and Suggestions for Authors
The manuscript concerns an important issue, which is the non-destructive study of sugar content in apples using the fluorescence hyperspectral imaging system (FHIS). This is a new issue, undertaken by few researchers. The authors demonstrate proficiency in using this method and have prepared an interesting manuscript. However, the manuscript requires refinement. In many places, there is unnecessary information or explanation, which sometimes reduces the readability and understanding of the text, e.g., the purpose of the study (lines 75-87) is quite confusing. Similarly, in the methodology, the inclusion of another apple variety (lines 86-89) is unnecessary:
"In this study, the apples were collected from Hanyuan apple in Sichuan Province to investigate the SC. The apple variety is the Red General, its fruits are slightly larger than those of the Red Fuji, and are uniform and tidy. Red General apples are available about 30-40 days earlier than Red Fuji, filling a gap in the market."
Similarly, it is rather unnecessary to describe a simple refractometer (Section 2.4).
So the purpose of the study is not well formulated. I would suggest trying to review and improve the entire manuscript more thoroughly, including conclusions. In general, there is a lack of presentation of the experimental data around which the studies and calculations were made. Data obtained by the non-destructive method are presented and the fitting models are described. However, as for the reviewer, the results of these studies are important to assess the reality and possibly the variability of the sugar content in the entire series of fruits.
Figure 1 – what source is this data based on?
Line 150: “Two to three drops (approx. 0.3 mL) of the apple juice were added to the detection area by a burette.” – The extract preparation methodology is not precise enough and may affect the conducted tests. Usually, a larger extract volume is taken, mixed and 2-3 drops are taken for determination using a refractometer. Taking the first 2-3 drops directly from squeezed apple tissue may differ from subsequent extract portions from the same fruit tissue.
242: „The curve of fluorescence hyperspectral image after ROI extraction by ENVI 5.3 showed different typical characteristic emission peaks, as shown in Fig. “ – No numbering. The active hyperlink takes to “Error! Reference source not found..” on line 260. This should be corrected.
Conclusions; Lines 405-407: “In this study, the feasibility of predicting the apples SC by fluorescence hyperspectral image system was studied. Non-destructive detection of the apples SC could be achieved by combining the FHIS with ML.” - I wonder if the tone of these effects can be used to predict sugar content, as stated in the conclusions. This may be a bit of an inaccurate word. After all, it was about possibly using a fluorescent hyperspectral imaging system (FHIS) for non-destructive testing of apple sugar content. We can, therefore, talk about the research accuracy (effectiveness) of this method compared to real conditions.
In any case, the research is very important and the described method has great potential for application on a wider scale.
I also wanted to emphasize the great knowledge and expertise of the authors of this manuscript in modeling and machine learning.
Comments on the Quality of English LanguageThe English could be improved to more clearly express the research.
Reviewer 2 Report
Comments and Suggestions for Authors
The topic is interesting. I suggested several additions and clarifications in the manuscript
